# Quality Standard for Rehabilitation of Young Deaf Children Receiving Cochlear Implants

**DOI:** 10.3390/medicina59071354

**Published:** 2023-07-24

**Authors:** Leo De Raeve, Marinela-Carmen Cumpăt, Aimée van Loo, Isabel Monteiro Costa, Maria Assunção Matos, João Canossa Dias, Cristian Mârțu, Bogdan Cavaleriu, Alois Gherguț, Alexandra Maftei, Ovidiu-Cristian Tudorean, Corina Butnaru, Roxana Șerban, Tatiana Meriacre, Luminița Rădulescu

**Affiliations:** 1Independent Information Center on CI’s (ONICI), 3520 Zonhoven, Belgium; leo.de.raeve@onici.be; 2Clinical Rehabilitation Hospital, 700661 Iasi, Romania; cristimartu@gmail.com (C.M.); cavaleriu@yahoo.com (B.C.); 3Department of Medical Specialties, University of Medicine and Pharmacy ”Grigore T. Popa”, 700115 Iasi, Romania; cmbutnaru@yahoo.com (C.B.); roxana_serban10@yahoo.com (R.Ș.); tatiana.meriacre.md@gmail.com (T.M.); lmradulescu@yahoo.com (L.R.); 4Prevention Center Zuyd, Zuyd University of Applied Science, Nieuw Eyckholt 300, 6419 DJ Heerlen, The Netherlands; aimee.vanloo@zuyd.nl; 5School of Health Sciences (ESSUA), University of Aveiro, 3810-193 Aveiro, Portugal; imonteiro@ua.pt (I.M.C.); maria.matos@ua.pt (M.A.M.); joao.canossa@ua.pt (J.C.D.); 6Center for Research in Health Technologies and Services (CINTESIS@RISE), 4200-450 Porto, Portugal; 7Faculty of Psychology and Education Sciences, Department of Education Sciences, “Alexandru Ioan Cuza” University of Iași, 11 Carol I Boulevard, 700506 Iasi, Romania; alois@uaic.ro (A.G.); psihologamaftei@gmail.com (A.M.); cristi.tudorean@gmail.com (O.-C.T.); 8Society of Otology and Cohlear Implant (SOIC), Str. Pantelimon Halipa nr 14, 700661 Iasi, Romania

**Keywords:** cochlear implants, language therapists, speech recognition, speech intelligibility, quality standards

## Abstract

*Background and objectives:* More and more children with severe-to-profound hearing loss are receiving cochlear implants (CIs) at an early age to improve their hearing and listening abilities, speech recognition, speech intelligibility, and other aspects of spoken language development. Despite this, the rehabilitation outcomes can be very heterogeneous in this population, not only because of issues related to surgery and fitting or the specific characteristics of the child with his/her additional disabilities but also because of huge differences in the quality of the support and rehabilitation offered by the therapist and the family. These quality standards for the rehabilitation of young deaf children receiving CIs are developed within the European KA202 Erasmus+ project “VOICE”—vocational education and training for speech and language therapists and parents for the rehabilitation of children with CIs, Ref. No.: 2020-1-RO01-KA202-080059. *Material and methods:* To develop these quality standards, we used the input from the face-to-face interviews of 11 local rehabilitation experts in CIs from the four partner countries of the project and the outcomes of the bibliographic analysis of 848 publications retrieved from six databases: Pub Med, Psych Info, CINAHL, Scopus, Eric, and Cochrane. Based on all this information, we created a first set of 32 quality standards over four domains: general, fitting, rehabilitation, and for professionals. Further on, the Delphi method was used by 18 international rehabilitation experts to discuss and agree on these standards. *Results:* The results from the literature analysis and the interviews show us that more than 90% of the consulted international experts agreed on 29 quality standards. They focus on different aspects of rehabilitation: the multidisciplinary team, their expertise and knowledge, important rehabilitation topics to focus on, and programming issues related to rehabilitation. *Conclusions:* These quality standards aim to optimize the activity of speech rehabilitation specialists so that they reach the optimal level of expertise. Also presented is the necessary equipment for the IC team to carry out the rehabilitation sessions in good conditions. This set of quality standards can be useful to ensure the appropriate postoperative care of these children. As a result, the rehabilitation process will be more relaxed, and therapists will have the opportunity to focus more on the specific needs of each child, with the provision of quality services, which will result in better results. This theme is particularly complex and dependent on multifactorial aspects of medicine, education, speech therapy, social work, and psychology that are very intricate and interdependent.

## 1. Introduction

The introduction of universal neonatal hearing screening made an earlier diagnosis of hearing loss in children possible. The benefits of early identification and intervention [1,2,3,4,5,6] are well known. Children fitted with hearing aids before the age of six months have a better outcome in terms of speech perception and speech production in the end, and they have a better chance to enter a mainstream educational setting [3,5,7].

Nowadays, more and more deaf children are receiving CIs, which results in a positive impact on their auditory perception [8,9] and language development [9,10,11,12]. These effects are even stronger in children who receive cochlear implantation early in life [10,11,13,14,15,16,17]. Yoshinago-Itano [3] concluded that the results of early cochlear implantation in deaf children, together with a high-quality rehabilitation program, may have similar results to those of children with mild-to-severe hearing loss wearing conventional hearing aids.

Furthermore, the WHO reports in the World Hearing Report [6] that timely intervention with cochlear implants leads to a better academic outcome and substantially improves the quality of life. In this World Hearing Report, the WHO also states that it is necessary that the rehabilitation of hearing loss is family-centered, and the approach to the deaf child is to answer to his individual and specific needs, of course, related to the resources available.

The importance of the content of the rehabilitation program and family support was also confirmed by Percy-Smith and colleagues [18], who found significant regional differences in the outcomes after CIs in Denmark. Although the cohorts that were studied and compared were matched in terms of age at implantation and hearing thresholds with CIs, the outcomes were significantly different in favor of those from East Denmark. They found that these differences arose mainly from the fact that the parents from West Denmark did not participate enough in the rehabilitation of their children and second from the fact that they used sign language to the detriment of spoken language. As a result of this study, a 3-year auditory verbal therapy (AVT) program has been set up since 2017 as part of the Danish healthcare system for children aged 0–5 years with severe bilateral hearing loss. The results of the program were significant: the majority of children acquired, after 3 years of AVT (which was less than 50%), age-equivalent spoken language, and through the process, 93% of parents were satisfied or very much satisfied with the program [19].

There are many studies on the benefits of CIs and the factors that influence the rehabilitation of these children. The factor recognized as the most important to obtain significant results is early implantation [20,21]. However, the diversity of the results obtained, even in conditions where all children are implanted early, must be considered. Children with CIs form a very diverse population with many variables to consider. Boons et al. [22] divided the predictive factors into three groups: child-related factors (aetiology, cognitive skills, and additional disabilities), auditory factors (age diagnosis, age of CI, unilateral or bilateral CI, and fitting), and environmental factors (multilingualism, communication mode, family environment, rehabilitation, and education). Rehabilitation appears as an essential factor in predicting the outcomes of these children [22,23,24], but it is not clear yet how to optimally involve parents and how much and what kind of support a child with one or two CIs needs.

Due to the huge differences in the service delivery models and intervention between countries, and even within the same country, between CI teams [18], we have created quality standards for the rehabilitation of children receiving CIs. The development of these standards is part of the European KA202 Erasmus+ project “VOICE”, vocational education and training for speech and language therapists and parents for the rehabilitation of children with CIs, Ref. No.: 2020-1-RO01-KA202-080059. Seven partners from four European countries are involved in this project: the Clinical Rehabilitation Hospital, Iasi, Romania; the Society of Otology and Cochlear Implants (SOIC), Romania; the University Alexandru Ioan Cuza of Iasi, Romania; the EuroEd Foundation, Romania; the Independent Information Center on Cochlear Implants (ONICI), Belgium; the University de Aveiro, Portugal; and Zuyd University of Applied Sciences, the Netherlands.

The main objective of our project is to train speech and language pathologists (SLPs) and parents to teach children with CIs to understand speech and to speak. In the scope of this project, we also identified quality standards for good practice to help service providers maintain and improve their standards and quality service provision for young deaf children receiving CIs.

Rehabilitation, according to the WHO [25], represents all the actions carried out to optimize the functioning and reduce the disability of people with health conditions in their interactions with their environment. These activities are carried out by specialized health services that help people to acquire, maintain, or improve, partially or totally, skills related to communication and the activities of daily living. Rehabilitation refers to the gaining of skills, abilities, or knowledge that may have been lost or compromised as a result of illness, injury, or the acquisition of a disability [26].

## 2. Materials and Methods

### 2.1. Interview Local Experts

To develop the quality standards, the authors started by interviewing 12 local experts in rehabilitation of young CI children. Each of the 6 partners interviewed, in person or online, 2 local experts in the field of rehabilitation, resulting in 12 experts from 4 countries (Belgium, the Netherlands, Portugal, and Romania). For personal reasons, we could not make the interview with one expert from the Netherlands, so the final total number of experts was 11 (Appendix B). The method used to select the local experts was nonprobability sampling. To assure the representativeness of our sample, we only included people from our target audience (SLTs who are experienced in working with cochlear-implanted children), knowing that representativeness is much more important than the size.

To have a standardized interview across the partners, we created 8 open questions to focus on during the interview (Appendix A):-What is your experience in rehabilitation of young deaf children with CI’s learning to speak and to develop spoken language?-Who (what kind of professionals/parents) should be involved in the rehabilitation process after CI in young deaf children?-What should be the role for the CI-team and what should we expect from local professionals? So, who is doing what?-What are important domains to focus on during the rehabilitation of young CI-children?-Should we involve the parents in the rehabilitation process? If so, Why + How + How often?-Do you prefer a specific rehabilitation program or approach for therapy? Which resources (books, publications) do you suggest for other therapists to become an expert too?-Do we have to monitor the listening, speech and spoken language development of these young CI-children? If so. Why and how often?-What frequency of therapy sessions do you suggest for these young CI-children and their families? How often? For how many minutes? For how many months/years?

The content of all these interviews was analyzed and structured following the 8 open questions above. We focused especially on common answers between the local experts. All this was summarized in a report, which is available from the project website https://voice-erasmus.eu (accessed on 12 January 2023)

### 2.2. Standards—Bibliographic Context Analysis

The topic itself is emergent because of the recent technological history, with few professionals involved and with procedures that are not consolidated yet.

To have input from latest scientific research, we performed a review of literature on rehabilitation of young children receiving CIs. The present methodological framework is based on the Preferred Reporting Items for Systematic Reviews and Meta-Analyses (PRISMA) guidelines to ensure replicability and transparency [27]. 

All studies at all levels of evidence were included: best practice, case studies, prospective and retrospective studies, cross-sectional and longitudinal studies, and (non)randomized control studies (RCSs). The articles had to be published in peer-reviewed journals or books in the English language between 2000 and 31 May 2021. The articles had to focus on CI children under age of 6 with bilateral hearing loss.

#### 2.2.1. Search Strategy

The literature search was conducted between 31 May 2021 and 31 August 2021 using a 4-word search (cochlear implant + children + rehabilitation + education) through the following 6 databases: PubMed, Eric, Cochrane, CINAHL, Scopus, and Psych Info.

Three authors (L.D.R., A.M., and T.M.) performed independent systematic title and abstract screening based on the predefined inclusion criteria. A.M. and T.M. first performed the same literature search using PubMed. Their outcomes were compared to each other (inter-reliability) and checked independently by the main author (L.D.R.). As the inter-reliability score was very high (>90%), A.M. further performed the search on Cochrane and Scopus, and T.M., together with L.D.R., performed the literature search using Eric, CINAHL, and Psych Info. Separate from the search through the 6 databases, PubMed (339), Psych info (203), CINAHL (189), Scopus (95), Eric (14), and Cochrane (8), there was also the possibility to add additional records identified through other sources (39). In total, we retrieved 848 publications.

#### 2.2.2. Level of Evidence

A bibliographic analysis was performed after more than 150 papers were evaluated from the point of view of the level of evidence, as defined by Lebwohl and colleagues [28], i.e., in a very brief presentation: level A refers to double-blind study; level B to prospective clinical trial with more than 20 subjects; level C to clinical study with less than 20 subjects, possibly case reports, or retrospective data analyses; level D to clinical study with less than 5 subjects; and level E to brief case-by-case presentations or expert opinions.

The level of evidence from all full-text articles was scored by 5 coauthors (L.D.R., A.v.L., I.M.C., M.A.M., and C.M.). All scores were extra-checked by the main author L.D.R., and in case of a different score (which exceptionally happened), they had a short discussion to come to a final score.

### 2.3. Quality Standards

The development of the quality standards involved a modified Delphi consensus process that was informed by outcome of the interviews of the local experts and the extensive analysis of the literature. Based on all this information, we created a list of 32 quality standards over 4 domains related to rehabilitation: (1) general, (2) fitting, (3) rehabilitation per se, and (4) standards for professionals.

An international group of world-renowned clinical experts in the field of rehabilitation of young children with CIs and with significant experience joined a Delphi consensus panel. All 6 partners suggested 3 international experts on which they have based the rehabilitation in their center. The aim was to develop a series of quality standards for rehabilitation of children using CIs.

The Delphi method was modified to include 2 rounds of email questionnaires. In the first round, we asked the 18 international experts (Appendix C) in the field of rehabilitation to give their remarks and suggestions concerning the 32 quality standards. After adapting the quality standards, we asked the international experts in the second round to agree or not agree with each quality standard.

The proposed standards on rehabilitation of deaf children receiving CIs offer optimal level of expertise necessary for staff working in field of speech therapy and present the facilities CI teams should have. They cover different aspects of rehabilitation, such as multidisciplinary, the necessary knowledge, the domains to focus on, CI programming, and so on.

## 3. Results

### 3.1. Interview Local Experts

The group of local experts consisted of 10 speech and language pathologists and 1 otolaryngologist, with an average experience of 13 years in the field of pediatric cochlear implantation. All mentioned that the CI team should be a multidisciplinary team, which consists of an otolaryngologist, audiologist, and speech and language pathologist as a minimum. Preferably, also a psychologist, social worker, teacher of the deaf, and a physical or occupational therapist should be included in the team. Parents should be seen as equal partners too, and nearly all the experts (8/11) suggested to use a family-centered approach.

The local experts also suggested that the CI team should be the coordinator of the rehabilitation/aftercare and that there should be good liaison with the local support team. The most important domains to focus on during the rehabilitation process are audition/hearing/wearing devices; communication/speech/spoken language (11/11); parent (care) support/coaching (8/11); cognitive development (7/11); social–emotional development (6/11); reading/school performance (4/11); motor skills/planning (4/11); and self-advocacy/identity (3/11).

Auditory verbal therapy (AVT) was mentioned by 7 out of 11 experts as the approach which fits best most children and their families, but on the other hand, the same experts stated there is not one approach that fits all.

Most local experts (9/11) also mentioned that during the rehabilitation process, all steps in the development of the child should be monitored regularly, especially hearing and speech and language development. Concerning the frequency of therapy, there was huge variability between the experts, but ideally, most of the experts (6/11) suggested one session of 1 h on a weekly basis. But it was also mentioned that the frequency of rehabilitation will depend on the child, the family, the distance (although it can also take place online), and the availability of support services.

### 3.2. Bibliographic Analysis

In total, 848 individual publications were identified (Figure 1), but after removing the duplicates, 790 were left. In the next step, we excluded 618 records because they did not fall within the inclusion criteria: the content of 295 publications was not about rehabilitation, 214 were not about CI children with a bilateral hearing loss, 99 were not about young children (<6 years of age), and 10 were not in the English language. Therefore, we had 172 full-text articles to assess for eligibility and excluded another 22 publications based on population (15), the same study sample (6), and missing relevant information (1).

Finally, we could include 150 publications in the qualitative synthesis, and for this purpose, we only kept the 66 publications with an A or B level of evidence and excluded those that scored C, D, or E. Of these 66 publications, only 2 publications [29,30] received an A score on their level of evidence, which illustrates how weak the level of evidence is, in general, in studies related to the rehabilitation of children with CIs.

Within the field of the rehabilitation of children with CIs, it is exceptional to find research based on double-blind studies. A good level of evidence is mostly shown by publications on prospective clinical trials on more than 20 subjects, lacking adequate peer group controls or another key facet of the design.

There is still no consensus on the most appropriate methodology for approaching the rehabilitation of children with CIs [31,32]. A recent systematic review still abounds in the same direction. Fitzpatrick and colleagues [32], Kaipa and Danser [33], and Demers and Bergeron [24] showed that more data are needed to decide on rehabilitation methodology. Most articles present a relatively low level of evidence. Therefore, there are not enough elements to rely on in choosing the approach for children receiving CIs, at least as far as we know.

### 3.3. Quality Standards

Based on the comments of the international experts during the first round, especially focusing on the overlap between some quality standards, we updated the quality standards and came to a new list of 29 quality standards. During the second round, we asked the same international experts to agree or not agree on the proposed standards. Finally, more than 90% of the international experts agreed on 29 quality standards over four domains: (1) general standards, (2) fitting, (3) rehabilitation, and (4) quality standards for staff, of which most of them contain several subcategories.

These standards for the rehabilitation of children with CIs can help health systems and, more specifically, CI teams and all those involved in the rehabilitation of children with CIs, to provide quality and State-of-the-Art care. We are absolutely convinced that by using these standards, they will improve the quality of their service delivery, and the result will be rehabilitation with better results.

## 4. Quality Standard for Rehabilitation of Young Children Receiving CIs (Appendix A)

### 4.1. General Quality Standards for Pediatric Cochlear Implantation in Relation to Rehabilitation

▪Providing a Child with a Cochlear Implant Requires a Dedicated Multi-Disciplinary Team Consisting at Minimum of an Otolaryngologist, Audiologist and a SLT Specialist
-Ideally there should also be a psychologist and social worker included in the team.-The multidisciplinary team should work inter professionally (not next to each other but together) and in close cooperation with the parents/carers.-The multidisciplinary team will liaise and work with the child’s local support team.▪The CI-Team of the Hospital Should Coordinate the Selection, Surgery, Fitting, Rehabilitation and after Care (Equipment Maintenance, Spare Materials)▪Parents/Educators/Professionals Need Balanced and Unbiased Up-to-Date Information about CI’s and the Fitting/Rehabilitation Process
-Ideally in their language.-Parents/educators/professionals should get appropriate counselling from the CI team and other professionals to have appropriate expectations from the cochlear implant, depending on several variables such as age at implantation or additional disabilities.-Parents/educators/professionals should have the opportunity to meet other families with CI children.-Parents/educators/professionals also need psychological support: taking care of their emotions and stress.▪Rehabilitation should Be Delivered by the CI-Team in Close Cooperation with A Local Expert (Team) in Listening and Spoken Language Development. (See Quality Standard # 29)▪Rehabilitation Is Not Possible without Parent/Family/Caregiver Involvement
-In case of parents or legal guardians are not able to be actively engaged in the child’s rehabilitation due to very low Social Economical Status, mental health matters, or cognitive delays, other family members or carers should be involved.-Professionals should use a child/family cantered approach.▪The Cochlear Implant Surgery Should Take Place as Soon as a Child Is Identified as a Candidate and Should Ideally Be Done by the Age of 12 Months or Sooner, Preferably under the Age of 36 Months, without Excluding Those Older Than 37 Months▪A Child with Bilateral Deafness Should Be Fitted Bilaterally with CI’s, Preferably before the Age of 18 Months
-We expect all countries to follow the principles and guidelines of the Joint Committee on Infant Hearing to have early hearing screening (before age 1 month), diagnosis (before 3 months) and start with rehabilitation (fitting hearing aids and early intervention) before 6 months of age.▪The CI-Team will Issue or Dispatch Replacements for Faulty External Equipment within Two Working Days
-There should be a written policy regarding who is responsible in the event of loss/damage and what spares can be provided as a matter of routine.▪Each Child’s Sound Processor Must Be up Graded Every 5 Years▪The National CI Program should Conduct and Publish Annual Audits and Comply with the Requirements of the Responsible National Authorities. Audits should Cover All Aspects Related to CI: Clinical Activity, Staff Expertise, Child Outcomes, Surgical Complications, Device Failures, and Child and Family/Caregiver Feedback on the Service Provided
-The audits should become freely available to interested parties.

### 4.2. Quality Standards on Fitting/Programming in Relation to Rehabilitation

▪The Fitting of the Sound Processor should Be Carried out by Qualified Paediatric Audiologist Preferably in Clinic, Face-to-Face Rather Than Remotely▪There should Be a Liaison between the Audiologist of the CI-Team and the Local Rehabilitation Expert/Local Support Team (and Vice Versa) to Exchange Information about the Progress of the Child’s Auditory Skills
-It is recommended that local professionals receives writen reports on the child’s auditory function-It is recommended that local professionals received written reports on the child’s auditory performance.▪Instructions on the Use of the Sound Processor Must Be Given to the Parent/Caregiver on or before the Day of Activation and should Be Repeated at Least Twice within the Six Months Following Activation
-This is within the role of the audiologist and the rehabilitation therapist.-Supporting materials on the handling, operating and care of the sound processor should be issued to the parent/carer.-The recommended use of assistive listening accessories (e.g.,: AudioStream, Mini mic, Roger) should be explained to the parent/carer by the CI team (see Quality standard # 1) before the CI surgery and the information reviewed after the activation.▪Appropriate Audiological, Standardized Speech Perception Tests and Functional Hearing Assessment (by Family/Other Professionals’ Questionnaire) should Be Performed at 6 Months Intervals to Enable Hearing to Be Monitored
-It is recommended to assess speech perception with standardized tests and a functional hearing questionnaire. Ideally: every 6 months in the first 2 years after the cochlear implant activation and then every year minimum of once a year.-Measuring speech perception of soft speech and speech in noise should begin after two years of CI use.-The results should be shared with the child’s parents/educators and local professionals.

### 4.3. Quality Standards on Rehabilitation of Young Children Receiving CI’s

▪Rehabilitation should Begin before Implantation and at the Latest Immediately after Initial Fitting, According to the Individual Needs of the Child
-Even if rehabilitation does not start until initial fitting, written material about the content of rehabilitation should be shared with the parent/carer well before initial fitting, so that they have a good idea of what is needed to promote an appropriate child’s listening and spoken language development.▪Parents/Educators/Professionals Are Considered and Valued as Equal Partners in the Rehabilitation Process of Their Child
-Parents/educators/professionals must have equal access to information on CI in their preferred language.-CI companies should make their brochures available in the preferred language of the parents/caregivers.▪Appropriate Measures should Be Performed Yearly (Ideally Every 6 Months) to Monitor Progress in Language, Communicational and Educational Outcomes the First 3 Years after Implantation
-Standardized assessments for typical hearing children should be used for comparisons.-Additional rehabilitation and/or referrals should take place where progress is slower than expected.▪A Diagnostic Coaching Approach to CI Rehabilitation Yields the Most Efficient and Best Benefit, Both to Children and to Parents/Educators/Professionals
-Additional needs should be identified as soon as possible, so rehabilitation and expectations can be adapted to the special needs. Additional specialists in other fields can be incorporated into the team to share their expertise.▪The Audiologist and Speech and Language Therapist Together with the Parents should Decide on the Frequency of Specialist Contact Sessions for Fitting and for Rehabilitation Based on the Individual Needs of the Child and Their Family▪As the Recommended Approach of Services Is Family-Centered, It Is Understood That Rehabilitation Therapy Sessions can Take Place Weekly or Fortnightly, Considering That Most Listening and Spoken Language Experience Will Occur at Home between the Sessions▪Children with CI’s should have Annually the Opportunity to Trial and Assess Assistive Listening Devices (FM-Systems, Bluetooth Accessories)▪Rehabilitation of Young CI-Children should Involve Collaboration between the CI Centre, Local Professionals, and Parents/Educators to Cover the Following Areas:
-Listening skills/functional listening/speech perception.-Speech intelligibility, voice quality and prosody.-Communication skills including repair strategies.-All aspects of language development.-Theory of Mind development.-Ability to troubleshoot and maintain external equipment.-Using assistive listening devices.-Music.-Literacy (reading).-Cognitive skills (Executive Functions).-Mainstream education (inclusion).-Advocacy.▪It Is Recommended That All Children to Receive, Based on Their Need, Listening and Spoken Language Therapy after Implantation, Even Those Who Benefit Little from CI and Who Are Anticipated to Still Be Sign Reliant
-Among all auditory-based early intervention approaches for children receiving CI, evidence-based practice has proven that an approach focusing on listening and spoken language has the most impact on the child’s speech perception skills and expressive spoken language development.-The decision to add signed support or sign language in the rehabilitation therapy will be discussed among parents and professionals so parents can make an informed decision.▪Rehabilitation Therapists and Parents/Educators will Collaboratively Generate Measurable and Appropriate Goals in All Areas of the Child’s Development (Auditory, Receptive, and Expressive Language, Speech, Cognition, and Social Skills), and Identify Ways to Integrate the Goals and Strategies to Achieve them in a Nurturing and Rich Language▪Music should Be Integrated in the Rehabilitation of Young Children Using CI’s, Particularly as a Home-Based Fun Activity Rather Than in a Formal Setting

### 4.4. Quality Standards for Professionals in Relation to Rehabilitation

▪Every Country should Have Training Opportunities for Professionals in the Various Communication Approaches (From Auditory Verbal to Sign Bilingualism) to Become An Expert in the Field of Rehabilitation and Education of CI-Children▪The Staff of the CI Team in the Hospital and Local Rehabilitation Therapists should Have the Knowledge and Expertise That Enables Them to Work Effectively with Children Wearing CI’s, Including Those with Additional Needs Than Their Hearing Loss
-Some very complex children may need a very specialized service.▪Rehabilitation of Young CI-Children should Be Carried out by An Expert in Promoting Listening, Speech and Spoken Language Development, in Managing the Technology and the Environment
-The therapist should also have expertise in coaching and counselling parents.-The therapist should also have expertise in a family-centered approach.▪The Expertise of the Rehabilitation Therapist/Rehabilitation Team should Include the Following Skills:
-Expertise and skills working with infants and very young children (for pediatric services).-Expertise in auditory development and listening skills.-Knowledge on how to manage the technology.-Knowledge on how to manage the acoustics of the environment and on how to address challenging listening situations (e.g., assistive listening devices).-Gain insight into the impact of hearing loss on a child’s overall development (eg mental health, language, speech, cognition, social and literacy) and how to support these skills.-Knowledge of communication support teams, i.e., speech to text or sign language interpreters.-Knowledge of audiology and assistive listening technology.-To get to know the culture and language of the deaf community.-Knowledge on how to coach/guide families.-Knowledge on inclusion of a CI-child (in education and in the local environment).

These standards for the rehabilitation of deaf children receiving CIs can be downloaded as a pdf file from the project website https://voice-erasmus.eu (accessed on 12 January 2023)

## 5. Discussion

Over the past two decades, there has been a great deal of research into the benefits of CIs and the search for the best way to support young CI children and their families. It is important to note that the overall level of evidence in the studies included in our literature analysis is low. Only two studies had a high level of evidence. Therefore, the level of available evidence does not seem to be sufficient to support the existence of differences in auditory and speech and language development between different rehabilitation approaches. This observation is consistent with the findings of recent systematic reviews, which concluded that more data are needed to be able to guide professionals and parents in their decision-making process [24,32,33].

The studies selected for the bibliographic analysis present some methodological limitations with the impact of the level of evidence. Due to the methodological design used in most studies, it is difficult to identify the rehabilitation factor and state that the approach led to an observed difference between groups.

Meanwhile, good prospective clinical trials with 20 or more subjects focusing on variables that predict good outcomes are needed. Ideally, but difficult from an ethical point of view, longitudinal prospective studies are needed, which include many subjects and a control group, in order to find the best rehabilitation techniques for the desired results. Parameters, such as the frequency and duration of the sessions, the content of the rehabilitation sessions, different neurobiological processes, sequential vs. simultaneous cochlear implantation, monaural versus binaural implantation, and various materials and listening conditions, must be evaluated.

There is an urgent need for quality standards on the rehabilitation of young children receiving CIs. To our knowledge, this is the first international consensus study to be published on quality standards for the rehabilitation of young CI children. Twenty-nine quality standards were developed and endorsed by the Delphi consensus panel. These quality standards will function best with an experienced team with knowledge and expertise in early implantation, fitting, rehabilitation, parent coaching, monitoring children, and the maintenance of the device and aftercare.

In addition, there is the challenge of ensuring the appropriate rehabilitation methods for each child with their individual needs. The results of recent and future neuroscience research on developmental critical periods, synaptic plasticity, auditory cortex changes after cochlear implantation, and auditory processing will help to better design the rehabilitation procedures after implantation. Better evaluations of therapy methods, applying the new developed tests that assess the progress of the implanted child (e.g., voice quality measurements when the test is standardized and available for all SLTs) as well as an increased availability of auditory support programs might help to deliver them more readily and widely in a variety of languages. These challenges reside amongst all professionals and parents dealing with implanted children on a daily basis.

The most discussed topics during our Delphi consensus panel were those related to standards of rehabilitation per se. For instance, one standard that was removed after heated debates was that “The paediatric CI may be only carried out if the parents are able to take responsibility for the child’s rehabilitation”. At a certain point, the standard became “Paediatric CI may be only carried out if there is rehabilitation available after CI”, and finally, the standard was eliminated. Another standard that incited discussions was related to the recommendation to use sign language by the SLTs during the rehabilitation sessions. The majority of experts emphasized that “Rehabilitation approach that do not include signs language appear more frequent associated with a better auditory speech and language development”. But there were voices who said that “For some children a signed supported approach to CI therapy yields the most benefit, both for children, parents and teacher”. The standard was eliminated by itself but has been added to Standard 4.3.9. as a possible support method. The last eliminated standard was a reference to the “continuously diagnostic coaching approach to CI therapy yields the most benefit, both for children and to parents and teachers”; the idea of closely monitoring the progress of children has already been mentioned in Standard 4.3.3.

But there were some interesting talks related to some other standards; for instance, the reference to the frequency and quality of the therapy gave rise to numerous comments: “the frequency…of sessions for rehabilitation must be based on the individual need of the child”, “it also depends on the needs of the local service professionals”, “it is not frequency or the number of hours important but the quality of the therapy”, “I think that it can depend on the number and the level of expertise of the local service professionals than on their (child’s) needs”, “I think both frequency and quality of the therapy are very important after paediatric CI”, and so on. The result of this topic is illustrated by Standards 4.3.5. and 4.3.6.

A strength of this study on the quality standards for the rehabilitation of deaf children is that they were developed based on good practice from local experts, the evidence identified in a robust analysis of the medical literature on the topic of rehabilitation, and the expert opinion of an international Delphi consensus panel with experience in the rehabilitation of young children receiving CIs. This approach is in line with the American Academy of Otolaryngology–Head and Neck Surgery Foundation methods [34] for the development of clinical consensus statements, resulting in evidence-based consensus statements that are in line with clinical experience.

A limitation of the study is the minimal representation of Asia, Africa, and South America on the Delphi consensus panel and the preponderance of local experts from one country (Romania). However, the presence of two international experts from a Spanish speaking country and the fact that in most American Latin countries, the official language is Spanish, we expect that the standards may apply there too. The same applies for Afrikaans, a language that has its origin in German and Dutch languages, as long as we have international experts from Germany and Holland. We cannot tell if the consensus is true for SLTs from Asia, but we expect it is.

Another limit lies in the diversity of tests used to quantify the auditory and speech and language outcomes, which makes it difficult to compare results directly from the included studies. Furthermore, most studies did not clearly define the approach used or the intervention setting (frequency, length, and duration) that children were involved in.

Future research should try to avoid these limitations to bring more robust data to the field of rehabilitation. As many studies were based on retrospective data or cross-sectional designs, an important element that should be considered by future researchers is using a study design that has fewer methodological limits. Focusing on prospective and longitudinal data and using a bigger sample size would allow better control of the studied variables, enhance the level of evidence, and allow researchers to match the groups on key variables to reduce their influence on the results.

## 6. Conclusions

To develop quality standards for the rehabilitation of deaf children receiving CIs, we used the input from the interviews of 11 local rehabilitation experts on CIs from the four partner countries involved in the VOICE project and the outcome of the analysis of 848 publications related to the theme and retrieved from six databases. The Delphi method approach was used by 18 international rehabilitation specialists in CI intervention to discuss and agree on these quality standards. Finally, >90% of the international experts agreed on 29 quality standards, of which most of them contain some subcategories.

Further research is needed to address the issue of the rehabilitation of young deaf children receiving CIs. Studies involving larger samples, matched groups, and well-controlled interventions are essential to isolate the intervention factor and be able to generalize findings.

Meanwhile, we have to focus on good practice which takes into consideration the specific needs of the child, their family, and the contact they are evolving in [23].

We believe that the guidelines for good practices presented here can act as a lever for the necessary studies, as they already indicate the conceptual field where future studies should take place and, therefore, help in the creation of evidence-informed approaches.

## Figures and Tables

**Figure 1 medicina-59-01354-f001:**
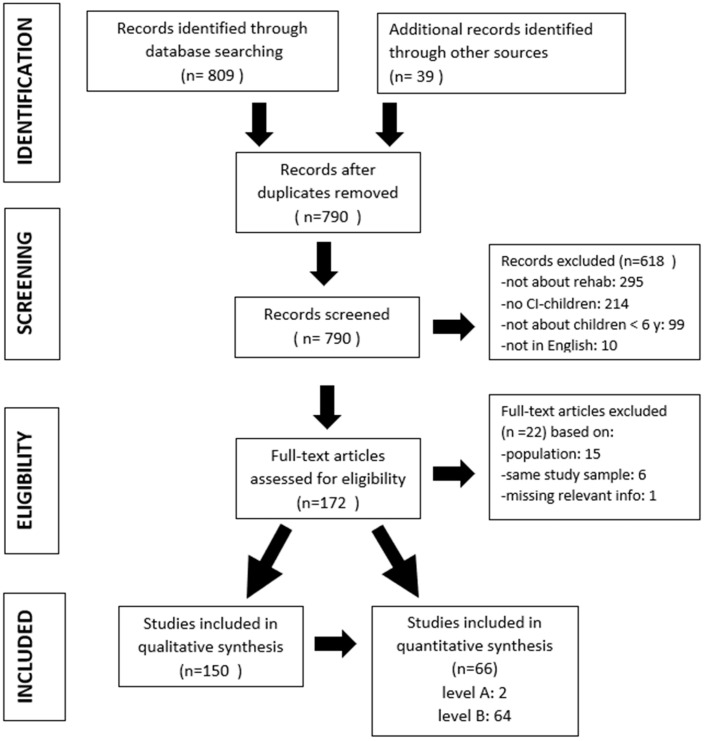
Flowchart of reviewed papers with exclusion and inclusion criteria.

## Data Availability

Not applicable.

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
