# Peer review of "Quality Standard for Rehabilitation of Young Deaf Children Receiving Cochlear Implants"

_medicina, 2023, doi:10.3390/medicina59071354_

Round 1

Reviewer 1 Report

Thank you for your efforts on this important topic. It is true there is diversity in rehabilitation for children with CI. However, I have a few comments on your paper

1.     Regarding the local experts that were interviewed to develop the standards, what method was used to select them, and were they enough in number and diverse in specialty to be a representative group to develop this consensus?

2.     The authors didn’t explain what led to the change in the number of standards from 32 to 29 after the 1st round.

3.     What are the detailed results of the 2nd round with international experts? It is better to present a table of the agreement and disagreement results, and any explanation for the disagreement standards

4.     It mentions limitations about the minimal representation of Asia, Africa, and South America, so these consensuses may not be appropriate for some countries

5.     From my understanding that local experts conclude 4 countries from Europe (Romania, Belgium, Portugal, and Netherlands). However, the international experts conclude some were from Romania, and a few of them were local experts as well, Is there any conflict in this manner?

6.     Line 480, Spelling correction for “Therefore”

.

Author Response

Dear reviewers of the “Quality standard for rehabilitation of young deaf children receiving cochlear implants” paper, Thank you all, first of all for your time and for the effort you put in reading and commenting on this article, and most of all, thank you for your suggestions.

Dear R1– thank you for your suggestions,

  1. Regarding the local experts that were interviewed to develop the standards, what method was used to select them, and were they enough in number and diverse in specialty to be a representative group to develop this consensus?

            In choosing the sample of local experts we used the method of nonprobability sampling including the people from our target audience (SLT that are working with STL children). In this way, we assured the representativeness of the sample because it is considered that this is much more important than the size (we added in the text this information – thank you!).

  1. 2.     The authors didn’t explain what led to the change in the number of standards from 32 to 29 after the 1st and 3.     What are the detailed results of the 2ndround with international experts? It is better to present a table of the agreement and disagreement results, and any explanation for the disagreement standards

            We added in the text the reasons for removing 3 standards out of 32 (please see lines 529 to 553), and also the experts’ comments on some topics.

  1. It mentions limitations about the minimal representation of Asia, Africa, and South America, so these consensuses may not be appropriate for some countries.

            We also added in the article some necessary clarifications with regard to the area of application of the consensus (lines 564-569).

  1. From my understanding that local experts conclude 4 countries from Europe (Romania, Belgium, Portugal, and Netherlands). However, the international experts conclude some were from Romania, and a few of them were local experts as well, Is there any conflict in this manner?

            We think that there is no conflict between presenting a problem or the way you solve the problem and discussing and finding, together with other experts, the best solution to that problem. This is an article about and for the CI team and especially for SLT – but all the team must be aware of the encounter problems during the rehabilitation because together we can find solutions. We think that in this kind of topic is the SLT the person who has the first say.

  1. Line 480, Spelling correction for “Therefore” solved.

Yours sincerely,

All the authors

Reviewer 2 Report

Thank you for sharing your work - it was a pleasure being involved in the review process.

The manuscript includes the new data obtained from the own study, which is not recommended to present in review type of publication. Meanwhile the results of the announced systematic review are absent, just the insufficient level of evidence of the selected studies is mentioned. The systematic reviews on the topic which are referenced to [31, 32] were performed at least 10 years earlier so systemizing and analysis of the recent studies are of the great value.

It would be more favorable to divide the presented material in two different articles – the systematic review and original article on development of the CI rehabilitation quality standard.

It is better to put "cochlear implants" in the article name instead of "CI" abbreviation.

The term "quality standard" may be added as a keyword.

Author Response

Dear reviewers of the “Quality standard for rehabilitation of young deaf children receiving cochlear implants” paper, Thank you all, first of all for your time and for the effort you put in reading and commenting on this article, and most of all, thank you for your suggestions.

Dear R2 – thank you for your suggestions,

The manuscript includes the new data obtained from the own study, which is not recommended to present in review type of publication. Meanwhile the results of the announced systematic review are absent, just the insufficient level of evidence of the selected studies is mentioned. The systematic reviews on the topic which are referenced to [31, 32] were performed at least 10 years earlier so systemizing and analysis of the recent studies are of the great value. It would be more favorable to divide the presented material in two different articles – the systematic review and original article on development of the CI rehabilitation quality standard.

            Indeed, using the term “systemic review” was misleading. The starting point of our study was an observation. We observed that there are no unitary principles for rehabilitation. We decided to ask the SLT about the methods they use in the rehabilitation of CI children. The answers were so different. Then we thought that it is important to check the literature so, we performed an extensive bibliographic analysis (that we incorrectly named it a systemic review). In the literature, we looked for answers to our questions. Based on answers from local experts, on knowledge from the literature analysis, and on the opinion of international experts we established baseline norms for rehabilitation of CI children – we call the Quality standards.

It is better to put "cochlear implants" in the article name instead of the "CI" abbreviation. – solved.

The term "quality standard" may be added as a keyword. – solved.

Yours sincerely,

All the authors

Author Response

Dear reviewers of the “Quality standard for rehabilitation of young deaf children receiving cochlear implants” paper, Thank you all, first of all for your time and for the effort you put in reading and commenting on this article, and most of all, thank you for your suggestions.

Dear R3 – thank you for your suggestions,

Dear authors, your study emphasizes the importance of a multidisciplinary team, early intervention, family-centered approach, and comprehensive support for optimal listening and spoken language development in pediatric CI users. As your aim was to provide standards of care, I aspected more details on diagnostic and evaluation instruments such as PRO measures, audiological test, phoniatric test and clinical evaluations (Frosolini A, Fantin F, Tundo I, Pessot N, Badin G, Bartolotta P, Vedovelli L, Marioni G, de Filippis C. Voice Parameters in Children With Cochlear Implants: A Systematic Review and Meta-Analysis. J Voice. 2023 Mar 1:S0892-1997(23)00021-8. doi: 10.1016/j.jvoice.2023.01.022. Epub ahead of print. PMID: 36868956), surgical standard (e.g. the choice of simultaneous implantation Uecker FC, Szczepek A, Olze H. Pediatric Bilateral Cochlear Implantation: Simultaneous Versus Sequential Surgery. Otol Neurotol. 2019 Apr;40(4):e454-e460. doi: 10.1097/MAO.0000000000002177. PMID: 30870380.). Overall I suggest, if possible, to implement more details regarding standard of measures and procedure at least in the discussion section. Moreover, a table that resembles the 28 quality standards should be helpful for the reader.

            We added in the text mentions about the suggestions from your comment.

            The entire field of rehabilitation of children with a cochlear implant is non-standardized, it is an emergent field, and it has a deep development character.

            We did not intend to impose norms or evaluation and diagnostic methods through this article.

            We proposed to present the findings of our pilot study, conclusions that emerged from discussions with local experts, through the analysis of a vast bibliography, and through discussions with international experts.

            Each member of the CI team plays a particular role.            This article is primarily addressed to speech therapists and naturally resulted from their desire to establish a consensus in order to have a clear basis on which to build the child's rehabilitation program according to the child’s needs.

            I think that through this approach progress can be faster and more consistent.

Yours sincerely,

All the authors

Round 2

Reviewer 2 Report

Dear authors, thank you for corrections. Would you be so pleased to include the list of 66 papers selected for bibliographic analysis in the supplement materials. In the field of cochlear implantation even the level B evidences can provide valuable information to other researches. The huge work that you have performed on the literature search will be useful for your readers.